# Occurrence and Distribution of Major Cassava Pests and Diseases in Cultivated Cassava Varieties in Western Kenya

**DOI:** 10.3390/v16091469

**Published:** 2024-09-15

**Authors:** Everlyne N. Wosula, Rudolph R. Shirima, Massoud Amour, Vincent W. Woyengo, Bonface M. Otunga, James P. Legg

**Affiliations:** 1International Institute of Tropical Agriculture (IITA-Tanzania), Dar es Salaam P.O. BOX 34441, Tanzania; e.wosula@cgiar.org (E.N.W.); a.massoud@cgiar.org (M.A.); j.legg@cgiar.org (J.P.L.); 2Kenya Agricultural Livestock Research Organization (KALRO), Kakamega P.O. Box 57811, Kenya; vincent.woyengo@kalro.org (V.W.W.); bonface.otunga@kalro.org (B.M.O.)

**Keywords:** *Bemisia tabaci*, whitefly, cassava green mite, cassava mosaic disease, cassava brown streak disease

## Abstract

Cassava is an important food crop in western Kenya, yet its production is challenged by pests and diseases that require routine monitoring to guide development and deployment of control strategies. Field surveys were conducted in 2022 and 2023 to determine the prevalence, incidence and severity of cassava mosaic disease (CMD) and cassava brown streak disease (CBSD), whitefly numbers and incidence of cassava green mite (CGM) in six counties of western Kenya. Details of the encountered cassava varieties were carefully recorded to determine the adoption of improved varieties. A total of 29 varieties were recorded, out of which 13 were improved, although the improved varieties were predominant in 60% of fields and the most widely grown variety was MM96/4271. The CMD incidence was higher in 2022 (26.4%) compared to 2023 (10.1%), although the proportion of CMD attributable to whitefly infection was greater (50.6%) in 2023 than in 2022 (18.0%). The CBSD incidence in 2022 was 6.4%, while in 2023 it was 4.1%. The CMD incidence was significantly lower (5.9%) for the improved varieties than it was for the local varieties (35.9%), although the CBSD incidence did not differ significantly between the improved (2.3%) and local varieties (9.7%). Cassava brown streak virus (CBSV) and Ugandan cassava brown streak virus (UCBSV) were both detected. Most infections were single CBSV infections (82.9%), followed by single UCBSV (34.3%) and coinfection with both viruses (16.7%). Whiteflies were more abundant in 2023, in which 28% of the fields had super-abundant populations of >100/plant, compared to 5% in 2022. KASP SNP genotyping designated 92.8% of the specimens as SSA-ECA for 2022, while it was 94.4% for 2023. The cassava green mite incidence was 65.4% in 2022 compared to 79.9% in 2023. This study demonstrates that cassava viruses, whiteflies and cassava green mites continue to be important constraints to cassava production in western Kenya, although the widespread cultivation of improved varieties is reducing the impact of cassava viruses. The more widespread application of high-quality seed delivery mechanisms could further enhance the management of these pests/diseases, coupled with wider application of IPM measures for whiteflies and mites.

## 1. Introduction

Cassava (*Manihot esculenta* Crantz) is one of the most important staple food crops in the world, supporting over 500 million people in sub-Saharan Africa (SSA) as a source of food and income [1,2]. The world’s production trend of this crop has been on the rise, with the current estimated production being 330 million tonnes in 2022, which represents an 80% increase in two decades compared to 182 million tonnes in 2002. In Africa, the production has doubled in two decades from 98 million in 2002 to 208 million tonnes in 2022, representing about 63% of the world’s production [1]. In Kenya (western and coastal regions), cassava is the second most important crop after maize [3,4,5]. Erratic weather patterns with reduced rainfall and increased dry spells due to climate change are a threat to food security, especially in sub-Saharan Africa, where the majority of people rely on rainfed agriculture. Cassava has emerged as a promising food security crop under the prevailing climate change threats due to its resilience and robustness allowing it to produce acceptable yields under marginal conditions associated with low soil fertility and low rainfall, and with minimal agronomic management practices compared to other crops [6]. Its ability to adapt and survive in different environments, combined with tolerance to prolonged dry spells, makes it one of the most important staple and food security crops in areas where production is constrained by soil stresses and civil strife [1,6,7]. Cassava production in SSA is progressively shifting from subsistence farming to commercial farming with the aim of providing raw materials for diverse products that include biofuel, starch, biopolymers, flour, and animal feed [8,9].

Despite Africa being the leading continent in terms of the production of cassava, the average yields are about 8 t/ha, which is extremely low compared to South America’s 14 t/ha and Asia’s 22 t/ha [1]. The low yield is attributed to poor agronomic practices and abiotic and biotic stresses. The most constraining factors in relation to cassava production in sub-Saharan Africa are two devasting viral diseases, cassava mosaic disease (CMD) and cassava brown streak disease (CBSD) [10,11,12]. CMD is caused by 11 species of cassava mosaic begomoviruses (CMBs) [12,13,14], while CBSD is caused by 2 species of cassava brown streak ipomoviruses (CBSIs), which are cassava brown streak virus (CBSV) and Ugandan cassava brown streak virus (UCBSV) [15,16,17]. The viruses that cause CMD and CBSD are spread through the propagation of infected planting material and transmission by cassava whiteflies—*Bemisia tabaci* (Genn.) (Aleyrodidae) [18,19]. CMD is prevalent in all cassava growing regions in SSA, while CBSD has been so far reported in East, Central and Southern Africa, with recent studies indicating the widespread occurrence of the disease in Uganda, Kenya, Malawi, Burundi, Rwanda, Zambia, Tanzania, Comoros Islands and some parts of the Democratic Republic of Congo [5,11,14,20,21,22,23,24,25,26]. Yield losses attributed to CBSD in SSA range between 30 and 40% and up to 70% for CMD [14]. In susceptible cassava varieties, CBSD is reported to cause up to 100% yield loss [27]. The most damaging effect of CBSD is root necrosis, causing root yield losses of up to 75% as the root is unmarketable or inedible in the most susceptible varieties [28]. The cause of the severe and rapidly expanding CBSD epidemics is yet to be identified; however, the introduction of the virus to mid-altitude areas and the presence of high whitefly populations are probable drivers of new CBSD outbreaks [11,29]. In a more recent report from Kenya, the CBSD foliar incidence ranged from 52.1 to 77.5% and the root necrosis incidence from 36.7 to 40% [30]. In a study conducted in 2013 in Kenya, CBSD resulted in root yield loss of 24.7%, translating to USD 1259.50 per hectare [30]. High CMD (71.4 to 100%) and CSBD (20 to 100%) incidences have been observed in western Kenya [31]. The combined losses from CMD and CBSD have been estimated at a value of USD one billion per year, threatening the livelihoods of smallholder households that depend on cassava as a staple food crop in sub-Saharan Africa [10,14].

*Bemisia tabaci* whitefly feeding on cassava can also damage plants, causing chlorotic mottling, twisting or curling, particularly in the upper leaves [32]. Large populations that develop early in the crop’s life reduce plant vigour and tuberous root sizes and cause stunted growth, leading to more than a 50% loss in yield [33]. A large whitefly population can also produce honeydew, which leads to the production of black sooty mould on lower leaves, reducing the photosynthetic ability of the plant, further contributing to yield losses [33,34]. However, the most significant economic threat is the spread of CMD and CBSD [29].

*Bemisia tabaci* comprises numerous mitotypes that have been identified based on sequences of the mitochondrial cytochrome oxidase I (mtCOI) gene [35,36]). The mitotypes found on cassava in Africa have been categorised into five major groups designated as sub-Saharan Africa (SSA1, SSA2, SSA3, SSA4, SSA5). SSA1 has five sub-groups: SSA1-SG1, SSA1-SG2, SSA1-SG3, SSA1-SG4, and SSA1-SG5 [29,37]). SSA1-SG1 is the most predominant mitotype in most cassava-growing regions of East and Central Africa, including Kenya and its neighbouring countries, Tanzania and Uganda [29,38]. Studies using SNP-genotyping through NextRAD sequencing identified six major genetic haplogroups (phylogenetic classification based on SNP genotyping) and showed that COI is not effective at distinguishing the major genetic groupings of cassava *B. tabaci* in Africa [38,39]. All the known mitotypes occurring on cassava were reassigned into the six SNP-based haplogroups: sub-Saharan Africa East and Central Africa (SSA-ECA), sub-Saharan Africa East and Southern Africa (SSA-ESA), sub-Saharan Africa Central Africa (SSA-CA), sub-Saharan Africa West Africa (SSA-WA), sub-Saharan Africa 2 (SSA2), and sub-Saharan Africa 4 (SSA4). A Kompetitive Allele-Specific PCR (KASP) assay has been developed to distinguish the six major SNP-based haplogroups [40].

The cassava green mite (CGM) *Mononychellus tanajoa* Bondar (syn = *M. progressivus*) (Acari: Tetranychidae) is a serious pest of cassava in sub-Saharan Africa that was accidentally introduced in the 1970s [41,42]. CGM causes damage through feeding on the undersides of young, emerging leaves, causing white to yellowish speckling, leaf and shoot deformation, and reduced size. Heavy infestations cause defoliation, beginning at the top of the plant, and often kill apical and lateral buds and shoots [42,43]. CGM damage is usually severe under dry conditions and high temperatures that favour rapid population build-up. However, under wet conditions and low temperature, the mite populations decrease, and plants tend to recover by producing new foliage [42]. The cassava yield loss due to CGM damage in Africa is 13–80%, depending on the cassava variety and prevailing climatic conditions [41,42,44]. Control of CGM relies on plant host resistance and biological control by various species of phytoseiid mites [42]. The successful introduction of *Typhlodromalus aripo* De Leon (Acari: Phytoseiidae) into cassava growing regions in Africa has contributed to significant control of CGM, with reductions of up to 45% [45,46,47]. Recent observations indicate that CGM could be re-emerging as a serious pest of cassava due to erratic weather patterns accompanied by prolonged dry periods or low rainfall intensity. These conditions could not only be favouring rapid population build-up of CGM but also hampering the survival and efficacy of phytoseiid predatory mites that are the main biological control measure [47]. The general objective of this study was to determine the status of major pests and diseases in cassava in six counties located in western Kenya, a leading region in terms of cassava production in the country, and based on that to propose management recommendations.

## 2. Materials and Methods

### 2.1. Survey Sites

Field surveys were conducted in six major cassava producing counties in western Kenya (Bungoma, Busia, Kakamega, Siaya, Homa Bay and Migori) during the short rainy season in October 2022 and repeated at the start of the long rainy season in March 2023. Ten fields with cassava plants three to six months old were sampled per county in each sampling season, giving a total of 120 fields sampled. The fields were selected along motorable roads, with a minimum inter-field separation distance of 5 km.

### 2.2. Whiteflies, Cassava Mosaic Disease, Cassava Brown Streak Disease and Cassava Variety Assessment

For each visited field, the cassava varieties, location information (county, district, location, village) and GPS coordinates were recorded. The assessment protocol was a slightly modified version of Sseruwagi et al. [48]. For each selected field, the predominant cassava variety was selected for assessments. Thirty plants were assessed per field by selecting plants at regular intervals along two diagonal axes, fifteen plants per axis. Adult whiteflies were counted from the top five leaves of the tallest shoot and the nymphs were counted from the 14th leaf of the 6th, 12th, 18th, 24th and 30th plants. Whiteflies were aspirated from randomly selected plants in each field, immobilised in 95% ethanol and preserved in 2 mL vials for molecular characterisation. The number of whiteflies collected per field was dependent on the abundance, ranging from 10 to 50 whiteflies per vial.

CMD was assessed using the 1–5 scoring scale, where 1 = cassava plant showing no leaf symptoms; 2 = mild distortion and mild chlorosis on the leaves; 3 = significant distortion and chlorosis on most leaves; 4 = extreme distortion and mosaic patterns on most leaves and general reduction of leaf size; and 5 = very severe mosaic symptoms on all leaves, accompanied by distortion and twisting, and severe leaf reduction of most leaves, accompanied by severe stunting of plants [48,49]. The CBSD foliar symptoms were scored using a scale of 1–5, where 1 = asymptomatic; 2 = mild vein yellowing with no streaks on stems; 3 = mild vein yellowing with mild streaks on stems; 4 = severe vein yellowing with severe streaks on stems; and 5 = severe and extensive vein yellowing, severe streaks, die-back and defoliation [50,51]. The leaves used for virus testing were sampled from 10 plants out of the 30 per field selected at regular intervals irrespective of whether the plants were symptomatic or asymptomatic. For plants that were symptomatic, leaves with virus symptoms were selected to increase the chances of molecular detection of the virus. Samples were preserved by pressing between two to three layers of blank newsprint sheets. The leaves were then left to dry and maintained at ambient temperature and moisture-free until required for further processing.

### 2.3. Cassava Brown Streak Ipomoviruses Detection by (RT) PCR Testing

The total nucleic acid (DNA and RNA) was extracted using a standard cetyltrimethyl ammonium bromide (CTAB) method [52] and the nucleic acids were re-suspended in nuclease-free PCR-grade water. CBSIs were detected using CBSV- or UCBSV-specific TaqMan assays [53,54] using an AriaMx Real-Time PCR System (Agilent technologies, Santa Clara, CA 95051 United States).

### 2.4. Genetic Identification of Bemisia tabaci

The whiteflies collected in this study were identified through COI sequencing, while those identified as cassava *B. tabaci* were further designated using KASP genotyping. Whitefly samples collected in a previous survey in 2017 from 8 counties (Busia, Bungoma, Homa Bay, Kakamega, Kisumu, Migori, Siaya, Vihiga) were included for comparison purposes to determine if there were any shifts in the mitotype distribution and proportion. DNA was extracted from 92, 102 and 120 single adult whiteflies for the years 2017, 2022 and 2023, respectively. A partial fragment of mitochondrial DNA cytochrome oxidase I (mtCOI) was amplified using one set of primers, 2195-Bt-F (5’-TGRTTTTTTGGTCATCCRGAAGT-3’) and C012-Bt-sh2-R (5’-TTTACTGCACTTTCTGCC-3’) [55]. The PCR reaction contained 1× QuickLoad Master Mix (New England Biolabs, UK), 1 mM MgCl_2_, 0.24 μM of each primer, 2 µL DNA, and sterile distilled water to achieve the desired reaction volume of 25 µL. PCR was carried out under the following conditions: 95 °C for 5 min for initial denaturation of the template DNA, followed by 35 cycles of (94 °C for 40 s, 56 °C for 30 s for annealing, and 72 °C for 90 s for extension), with a final extension at 72 °C for 10 min. The PCR products were run on a 1% (*w*/*v*) agarose gel in 1× TAE buffer stained with GelRed. The DNA bands were visualised while using a Gel Doc XR+ Gel Documentation System and only samples with intact bands of the expected size (867 bp) were selected for sequencing. The PCR products were sent to Psomagen Inc. (Rockville, Maryland, United States) for purification and direct PCR sequencing. The DNA sequences were manually edited using Ridom Trace Edit v1.1.0 (Ridom GmbH., Würzburg, Germany). The sequences were assembled into contigs using CLC Main Workbench 7.0.2 (QIAGEN, Aarhus, Denmark). Multiple alignment of the edited sequences was performed using ClustalW in Molecular Evolutionary Genetics Analysis software (MEGA version 7.0.26) [56]. Construction of a maximum likelihood phylogenetic tree was performed using MEGA with 1000 bootstrap replicates. The sequences were blasted using GenBank’s (NCBI) Blastn and selected reference sequences with 99% to 100% identity to our COI sequences were included in the phylogenetic tree for comparison with previously published mitotypes.

The cassava *B. tabaci* samples used to generate the COI phylogenetic tree for 2017 and 2023, and additional new samples (not diagnosed with COI for 2022 and 2023), were tested using the KASP diagnostic with a set of six primers (BTS99-319, BTS22-762, BTS141, BTS55-473, BTS613, and BTS46203) [40]. The total number of samples tested was 82 (2017), 111 (2022) and 232 (2023). Conventional primers were used to generate the PCR products of genome portions containing the target SNPs and the PCR products were then used as DNA template in the KASP genotyping [40]. The KASP reaction mixture (10 µL) contained 5 µL 2× KASP master mix, 0.14 µL KASP primer assay mix and 5 µL DNA template (1 µL of PCR product/DNA extract + 4 µL of sterile water). KASP genotyping was performed in a Stratagene MX 3000P qPCR system (Agilent Technologies, Santa Clara, California, United States). The following cycling conditions were used: Stage 1: 30 °C 60 s (pre-read); Stage 2: 94 °C for 15 min hot-start Taq activation (1 cycle); Stage 3: 94 °C for 20 s, 61 °C (61 °C decreasing 0.6 °C per cycle to achieve a final annealing/extension temperature of 55 °C) for 60 s (10 cycles); Stage 4: 94 °C for 20 s, 55 °C for 60 s (29 cycles); Stage 5: 94 °C for 20 s, 57 °C for 60 s (3 cycles); Stage 6: 37 °C for 60 s (1 cycle, cooling) followed by an end-point fluorescent read. These conditions were used for four primers (BTS99-319, BTS22-762, BTS55-473, and BTS141), while Stage 3: 94 °C for 20 s, 68 °C (68 °C decreasing 0.6 °C per cycle to achieve a final annealing/extension temperature of 62 °C) was used for two primers, BTS613 and BTS46-203. The quality of the genotyping cluster plots was visually assessed and only samples in distinct clusters with the respective positive controls were considered for manual SNP calling using the MxPro software incorporated into the Stratagene MX 3000P unit.

### 2.5. Cassava Green Mite Incidence

CGM damage was scored based on a scale of 1–5, where 1 = no obvious symptoms; 2 = moderate damage, no reduction in leaf size, scattered chlorotic spots on young leaves, 1–2 spots/cm; 3 = severe chlorotic symptoms, light reduction in leaf size, stunted shoot, 5–10 spots/cm; 4 = severe chlorotic symptoms and leaf size of young leaves severely reduced; and 5 = tips of affected plants defoliated, resulting in a candle stick appearance of shoot tips [57].

## 3. Results

### 3.1. Cassava Variety Distribution

The farmers’ fields sampled during the 2022 season had cassava fields ranging between 60 and 3000 m^2^, with an average size of 664 m^2^, while those for the 2023 season cassava fields ranged between 75 and 15,000 m^2^, with an average size of 979 m^2^. The altitude range of the sampled farms was 1138 to 1629 m a.s.l. for both seasons. In the 60 fields that were sampled in 2022, there were 21 different varieties that were listed as variety 1 (predominant and selected for data collection). Thirty-three fields (55%) had a second variety, 11 (18%) had a third variety and four (7%) had a fourth variety. Improved variety MM96/4271 (released in Uganda as NASE14) was the most frequent, being recorded in all the six counties in a total of 20 out of the 60 fields sampled. The second most frequent variety was Bumba—a local landrace—which was present in six fields (10%) in two counties. Of the remaining 19 varieties, four were present in two counties, while the remaining 15 were present in a single county (Table 1). In the 60 fields that were sampled during the 2023 season, there were 18 varieties that were listed as variety 1 (predominant). Thirty-six fields (60%) had a second variety, nine (15%) had a third variety and four (7%) had a fourth variety. Variety MM96/4271 was the most predominant, being recorded in all six counties in a total of 24 out of the 60 fields sampled. The second most common variety was MM95/0183, which was present in five fields in two counties. Of the remaining 16 varieties, 1 was present in two counties, while the remaining 15 were present in a single county (Table 2). The combined variety/landrace data for 2022 and 2023 show the top five varieties/landraces to be Bumba, Kamisi, MM95/0183, MM96/4271 and Migyera (TMS 30572/NASE3). MM96/4271 was the most predominant variety found in all the six counties, especially in Busia and Bungoma (Figure 1). The combined data for 2022 and 2023 show there were a total of 29 varieties recorded from 117 fields, while 3 fields had unknown varieties. Out of the 29 varieties, 13 were improved, accounting for 45%, and these were found in 72 fields, comprising 60% of the fields. The improved variety MM96/4271 was found in 43 fields, accounting for 36% of all the fields. Local landraces were found in 45 fields, equivalent to 37.5% of the total, while unknown varieties in three fields accounted for 2.5%.

### 3.2. Cassava Mosaic Disease

The incidence of CMD in 2022 was 26.4%, while the average severity was 2.7 and the prevalence was 68%. The incidence was calculated as the proportion of plants with symptoms, while the prevalence was the proportion of fields with symptoms. In fields where the CMD infection types were scored, the incidence of whitefly infection was 4.4%, compared with an average cutting infection incidence of 20.1%. The proportion of CMD-infected plants resulting from whitefly transmission was 18% (i.e., 4.4% whitefly incidence divided by 24.5% combined incidence for plants that were scored as either whitefly infected or cutting infected). The most affected varieties with incidences >50% were mostly local landraces (Magana, Nyakatanegi, Nyanjaga, Serere, Bumba, Fumba Chai, Yellow, Adhiambo Lera, Sudhe). The least affected varieties with <10% incidence were mostly improved (MM96/4271, MM96/5280, MM96/7151, MM95/0183, Migyera) (Table 1). Bungoma, with four varieties that were all improved, had a CMD incidence of 4–30%, with an average of 7.6%, severity 2.5–3.5, with an average of 2.7, and a prevalence of 70%. Busia, with five varieties, had a CMD incidence of 0–80%, with an average of 23.7%, severity 3.3–3.7, with an average of 3.4, and a prevalence of 60%. For the four varieties encountered in Homa Bay, the CMD incidence ranged from 0 to 62%, with an average of 24.6%, severity 2.9–3.9, with an average of 3.2, and a prevalence of 70%. The CMD incidence ranged from 0 to 93% in Kakamega, where six varieties were recorded, having an overall average of 42.6%, severity scores 2.8–3.1, with an average of 3.0, and a prevalence of 70%. Migori, with five varieties, had a CMD incidence of 0–100%, with an average of 24.0%, severity 3.8–4.0, with an average of 3.9, and a prevalence of 60%. Siaya, with seven varieties, had a CMD incidence of 0–60%, with an average of 35.7%, severity 3.1–4.0, with an average of 3.6, and a prevalence of 80% (Table 1). The county average CMD incidence was moderate to high in Busia, Homa Bay, Migori Kakamega and Siaya, in the range of 20–50%, while Bungoma had low incidence of 0–10% (Figure 2A).

Surveys during the long rainy season in March 2023 showed an average CMD incidence of 10.1%, average severity score of 3.0 and average prevalence of 60%. Out of the 10.1%, 4.4% was whitefly infection and 4.3% was cutting infection (not all the fields were scored as whitefly or cutting infection due to missing lower leaves). The proportion of CMD-infected plants resulting from whitefly transmission was 50.6% (i.e., 4.4% whitefly incidence divided by 8.7% combined incidence for plants that were scored as either whitefly infected or cutting infected). An incidence of CMD > 50% was recorded in only two varieties, Adhiambo Lera and “unknown”. The least affected varieties with <10% incidence were mostly improved (MM96/4271, MM96/0686, MM98/3567, MM95/0183, Migyera) and four local landraces (Nylon, Nyanchagi, Kamisi and Sudhe (Table 2). The CMD incidence was moderate in Homa Bay and Siaya (10–20%) but low in Bungoma, Kakamega, Migori and Siaya (0–10%) (Figure 2C).

Combining the CMD incidence for 2022 and 2023 across the counties, Siaya recorded the highest incidence of 26.5%, followed by Kakamega (25%), Homa Bay (22.2%), Busia (15.8%), Migori (15.7%) and Bungoma (5.4%). The CMD incidence in the improved varieties was 5.9%, with the most predominant MM96/4271 having an average of 3.5%, which was significantly lower (*p* < 0.0001) compared to the local landraces, which had an average of 35.9.0%. The combined severity score in the local landraces was 3.2, while for the improved varieties, it was 3.1, with the most predominant MM96/4271 having a severity score of 2.8.

### 3.3. Cassava Brown Streak Disease Incidence

In the 2022 survey, the average CBSD leaf incidence was 6.4%, while the average severity was 2.6 and average prevalence was 20%. The most affected varieties, with an incidence of >30%, were Red local, Magana, Nyakatanegi and MM96/4271. The remaining 17 varieties/landraces had an incidence of <5%, with the majority showing no CBSD symptoms. In Bungoma, out of the four varieties that were present, only MM96/4271 showed CBSD symptoms. The CBSD incidence was moderate in Busia and Homa Bay (10–20%) but low in Bungoma, Migori and Siaya (0–10%) (Table 1). No symptoms were observed in Kakamega (Figure 2B).

In the 2023 survey, the CBSD incidence was 4.1%, while the severity averaged 2.7 and there was a CBSD prevalence of 12%. The most affected varieties were Nyakatanegi (83%), MM96/4271 (6–25%) and an unknown variety (13%) (Table 2). Homa Bay was the most affected, with CBSD recorded in three out of the four varieties. CBSD symptoms were not recorded in Kakamega, Migori or Siaya counties (Table 2). The CBSD incidence was moderate in Homa Bay (10–20%) but low in Bungoma and Busia (0–10%), and no symptoms were observed in Kakamega, Migori and Siaya (Figure 2D).

When combining the CBSD incidence for 2022 and 2023 for each of the counties, Homa Bay had the highest incidence of 15.0%, followed by Busia (7.7%), Siaya (6.0%), Bungoma (2.7%), and Migori (0.2%). The CBSD incidence in the improved varieties was 2.3%, with the most predominant MM96/4271 having an average of 6.3%, while the local landraces had an average of 9.7%. Statistically, there were no significant differences in the CBSD incidence between the improved varieties and local landraces (*p* > 0.05).

### 3.4. Bemisia tabaci Adult Whiteflies and Nymphs

In 2022, the average adult whitefly abundance was 97, while that for nymphs was 78; however, excluding four fields with variety MM96/2480, which had extremely high numbers, reduced the average to nine adults and 31 nymphs (Table 1). The average adult whitefly numbers for counties ranged from 2 to 352: abundance was greatest in Homa Bay and least in Bungoma. Varieties that had an average of < 2 adult whiteflies and nymphs were MM96/5280, MH95/0183, Magana, Merry-go-round, Sudhe and Serere, whilst varieties with the greatest abundance of adults were MM96/2480 (1335), MM96/4271 (24) and Kamisi (34.6) (Table 1).

In 2023, the average number of whiteflies across the counties was 85, while for nymphs it was 93 (Table 2). The adult whitefly number was greatest in Bungoma and least in Homa Bay. The whitefly adult abundance was greatest for varieties MM98/3567 (282.1) and TMS I92/0067 (123.3) but averaged less than 5 and 2 for MM96/0686 and Sudhe (Table 2).

The combined 2022 and 2023 whitefly numbers on the improved varieties had an average of 63 adults and 66 nymphs (excluding four fields record on MM96/2480, which had an extremely high average number of adult whiteflies that was ~23 times the average in the improved varieties). The most predominant variety, MM96/4271, recorded an average of 74 adult whiteflies and 77 nymphs. The local landraces had an average of 26 adults and 58 nymphs, which was 58% and 12% fewer, respectively, compared to the improved varieties.

### 3.5. Real-Time RT-qPCR Testing Results for Cassava Brown Streak Ipomoviruses in Cassava Leaves—October 2022

The leaf samples that were collected in 2022 were tested for the presence of CBSV and UCBSV. CBSIs were detected in 86.5% of the samples that were scored as having CBSD, with the remaining 13.5% testing negative. CBSV was detected in 29 out of 35 samples that tested positive for CBSIs (82.9%) from five counties, while UCBSV was detected in 12 out of 35 samples (34.3%) in two counties (Homa Bay and Siaya) (Table 3). Asymptomatic plants collected across the six counties were tested for CBSIs, where 16% (86 out 556) were positive, ranging from 1% (Kakamega) to 34.5% (Bungoma). CBSV was detected in 88.4% of these cases (76 out of 86), while UCBSV was detected in 14% (12 out of 86) (Table 3). Considering the symptomatic plants tested from Homa Bay and Siaya, 25% were coinfected with CBSV and UCBSV, while for the asymptomatic plants, 9.2% were infected with both viruses.

### 3.6. Genetic Identification of Bemisia tabaci

Of the 92 (2017), 102 (2022) and 120 (2023) whitefly samples mtCOI sequenced, 73 (2017), 73 (2022) and 112 (2023) produced quality sequences. The sequences were combined with *B. tabaci* reference sequences from the GenBank to generate a Maximum Likelihood phylogenetic tree. The *B. tabaci* whitefly mitotypes for 2017 comprised SSA1-SG1 (64.4%), SSA1-SG2 (5.5%), SSA1-SG1/SG2 (1.4%) and SSA2 (28.8%); for 2022 comprised SSA1-SG1 (71.2%), SSA1-SG2 (27.4%), SSA1-SG1/SG2 and SSA2 (1.4%); and for 2023 comprised SSA1-SG1 (69.6%), SSA1-SG2 (24.1%) and SSA2 (4.5%). *B. afer* was detected only in 2023, accounting for 1.8% of the whiteflies sequenced (Table 4). In 2017, SSA1-SG1 was the predominant mitotype and present in all eight counties that were sampled. SSA2 was found in six counties (Bungoma, Homa Bay, Kakamega, Kisumu, Siaya, Vihiga), SSA1-SG2 in two counties (Busia, Siaya) and a single specimen of SSA1-SG1/SG2 was found in Vihiga (Figure 3A). In 2022, SSA1-SG1 was again the most abundant mitotype, followed by SSA1-SG2, and these were present in all six counties. SSA2 had only a single whitefly that was found in Siaya (Figure 3B). In 2023, SSA1-SG1 was still the most abundant mitotype, followed by SSA1-SG2, and these were found in counties. SSA2 was detected in three counties of Homa Bay, Kakamega and Migori (Figure 3C). SSA2 was the second most widespread mitotype after SSA1-SG1 in 2017, but in 2022 and 2023, the frequency of occurrence of SSA2 was very low (Figure 3A–C).

KASP SNP genotyping showed that two haplogroups (SSA-ECA and SSA2) were present, although some samples were not precisely designated into the six known SNP haplogroups (SSA-ECA, SS-WA, SSA-ESA, SSA-CA, SSA2, SSA4). SSA-ECA was the most widely distributed, accounting for 72.0% (2017), 92.8% (2022) and 94.4% (2023) of samples, while SSA2 accounted for 15.8%, 1.8% and 4.7% for the three years, respectively. A few samples were designated as others: 12.2%, 5.4% and 0.9% for 2017, 2022 and 2023, respectively (Table 5). The distribution for 2017, which had both SSA-ECA and 15.8% SSA2, shows that SSA-ECA is widespread across all six counties, while the SSA2 samples that were successfully genotyped were collected from Kisumu and Siaya counties (Figure 3D). The SNP haplogroups maps for 2022 and 2023 are not presented because they are over 92% SSA-ECA.

### 3.7. Cassava Green Mite Incidence

The average cassava green mite (CGM) incidence across the six counties in 2022 was 65.4%, with a severity score of 2.7. The most affected county was Migori with 82.7% and the least affected was Siaya with 49.6%. The CGM incidence in 2023 was 79.9%, with a severity score of 2.7. The most affected counties were Busia and Kakamega with 91.3% and the least affected was Migori with 64.0% incidence (Table 6). The CGM prevalence was 100% with the exception in 2022 of Kakamega (90%) and Siaya (80%) and in 2023 of Homa Bay (80%) and Siaya (90%). The average prevalence for both years was 95% (Table 6). In 2022, only three counties, Homa Bay, Kakamega and Siaya, had an incidence in the range of 30–60%, while Busia Bungoma and Migori had >60% (Figure 4A). In 2023, all the counties had an incidence >60% (Figure 4B). Variety MH95/0183 was the most affected, with a mean severity score of 3.3 and incidence of 83.3%, while the least affected was Bumba, with a severity score of 2.3 and incidence of 53.3%. In 2023, the reverse occurred, with Bumba as the most affected variety with a severity score of 3.2 and incidence of 98.3% while MH95/0183 had a lower severity score of 2.7 and incidence of 49.3%. MM96/4271, which was the predominant variety, had a severity score of 2.4 and incidences from 83.3 to 77.9% in 2022 and 2023 (Table 7).

## 4. Discussion

Field surveys in cassava production regions are critical to establish the occurrence of pests and diseases, and to determine the extent of the adoption of new and improved cassava varieties with resistance or tolerance to these biotic stressors. This study presents findings on cassava varieties, whitefly *B. tabaci*, cassava virus diseases and cassava green mite occurrence in six cassava growing counties in western Kenya. Cassava in western Kenya is mostly cultivated on a small scale, with average field area of ~0.08 hectares for a single predominant variety in the surveyed fields. The region has diverse cassava varieties, and most farmers (58%) had two varieties, while another 17% had three varieties. This indicates that the 29 varieties encountered in this study could just be a portion of what exists in farmers’ fields. Other studies have reported varieties that were not recorded in this study. For example, of the 18 varieties that were recorded in Migori [30], only 3 (MH95/0183, Migyera, Sudhe) were encountered in the current study. A high proportion (60%) of fields surveyed in the two years of the current study had improved varieties, and MM96/4271 was recorded across all six counties and in 36% of fields surveyed. MM96/4271 (NASE 14) is resistant to CMD and tolerant to CBSD [58,59]. It also has a high dry matter content, low cyanide content and good culinary qualities, factors that could have favoured its widespread adoption in western Kenya [58]. Another commonly cultivated variety in the surveyed area was Migyera (=TMS30572 = NASE3), which is also considered tolerant to CMD and CBSD [58,59,60]. Migyera was found to have only CBSV infection, while susceptible varieties had a mixture of both CBSV and UCBSV [59]. Both MM96/4271 and Migyera were recommended to farmers in Uganda as being CBSD tolerant because of the low virus incidence [59,61]. In addition, MM96/4271 was selected as the most preferred variety in a farmer participatory trial in West Nile, Uganda, where no CMD and CBSD incidence was recorded compared to 82% and 22.5%, respectively, in local landraces. It was also reported to have moderate CGM damage compared to local landraces [62]. The high proportion of farmers’ fields with improved varieties in western Kenya indicates farmers’ willingness to adopt these varieties with time, as long they have preferred attributes; for example, for the period of 1998–2001, the proportion of farmers’ fields with CMD-resistant varieties increased from 17 to 35% in Uganda [10]. Improved CMD-resistant varieties were introduced into Kenya in the 1990s as part of a region-wide programme to tackle the spreading pandemic of severe CMD [10,63,64]. Previous studies reported a high preference for local landraces by farmers in Migori in Kenya, and improved varieties were only being cultivated on 17.9% of the surveyed fields [30]. In Rwanda, a survey carried out in 2007 reported local landraces accounted for 83% of the varieties in farmers’ fields [65]. Generally, local landraces are susceptible to CMD, CBSD and CGM compared to improved varieties [10,65,66], although they often have specific quality traits that are preferred by farmers. In most cases, as observed in this study, farmers grow more than one variety, and in many cases, these are mixtures of both improved and local varieties and at different growth stages. These adjacent fields are usually the source of virus inoculum for infection where clean cassava cuttings are planted [65,67]

The high prevalence of CMD (64%) is an indication that the disease continues to be widespread in western Kenya, albeit at lower levels than in recent history. The average incidence was variable depending on the season of the survey, county, and cassava variety. The average incidence across counties and varieties was higher in 2022 (26.4%) compared to 2023 (10.1%). In 2022, all the counties had incidences above 10% except Bungoma, which had 8%. However, in 2023, four counties had an incidence of below 10%, with only Homa Bay and Siaya having higher incidences of 20% and 17%, respectively. The combined data for both years showed that Bungoma had the lowest CMD incidence of 5%, followed by Migori at 15%, while the other four counties recorded 16–27%. The CMD incidence differences were not statistically significant between counties (*p* > 0.05). The mean CMD incidence of 26.4% in 2022 is comparable to the 33% previously reported in Migori [30], 32% in the Comoros Islands [26], 33% in Rwanda [65], and 27% in Tanzania [68]. However, this was lower than the 52% reported in Zambia [69] and 84% reported in Benin [70]. More significantly, however, the current levels of CMD incidence in western Kenya are much lower than they were at the time of the severe CMD pandemic in the late 1990s, where the CMD incidence was 63% [10,63]. This appears to indicate a long-term impact of the introduction of CMD-resistant varieties. This observation is supported by the current study, as the CMD incidence was lower in the improved varieties (5.9%) compared to the local (35.9%). Furthermore, the most widely grown variety at the present time in western Kenya is MM96/4271, which had one of the lowest incidences of all (3.5%). A low incidence of CMD in the improved varieties compared to the local landraces has been reported in several studies; for example, in Migori, the incidence in the improved varieties was 18% compared to 47% in the local varieties [30], whilst in Uganda, there are reports of the complete absence of CMD in improved varieties compared with 82% incidence in local varieties [62].

CBSD, by contrast, had a low prevalence (16%) in western Kenya. The incidences of 6.4% in 2022 and 4.1% in 2023 were similarly low. The combined average incidence of 5.3% is very low compared to previous studies; for example, 42% in Migori [30], 23% in Uganda [62], 42% in the Comoros Islands [26], 21% in coastal Kenya [5], 32% and 35% in Tanzania [71,72], and 12% in the Democratic Republic of Congo [25]. The combined year incidence of CBSD was variable across counties at 0–23%, which could be attributed to factors such as the cassava variety, prevailing climatic conditions, health status of planting material, infection status of cassava plants in neighbouring farms, or age of the crop. This variability is reported in other studies; for example, in the coast region of Kenya, the incidence was 11–28% across three counties [5], Tanzania 0–98% [71,72], and Comoros Islands 30–49% [26]. The incidence in local landraces was higher (9.7%) than for improved varieties (2.3%), although the predominant variety MM96/4271 had an average CBSD foliar incidence of 6.3%, which was higher than other improved varieties. A lower incidence of CBSD in improved compared to local varieties has been reported elsewhere; for example, in Migori 27% (improved) versus 57% (local) [30], and in Uganda, 23% versus 82% [62].

Real-time quantitative RT-qPCR for CBSIs detected both CBSV and UCBV in samples collected in 2022. CBSV was the most commonly detected CBSI (82.9%), compared to 34.3% for UCBSV. These findings are comparable to previous studies that have reported a higher incidence of CBSV compared to UCBSV; for example, 74% versus 34% in the DRC [25], and 59% versus 54% in Kenya [5]. However, this contrasts with a study in Tanzania, where UCBSV (81%) was more frequent than CBSV (15%) for recycled planting material and 14% versus 2.6% for initially virus-free planting material [73]. The CBSV and UCBSV coinfection rate of 16.7% in this study is slightly higher than in previous studies that recorded 5% [60] and 8.4% [5]. The occurrence in this study of some plants with CBSD symptoms testing negative and asymptomatic plants testing positive for CBSIs has been reported in other studies and could be attributed to the cryptic nature of CBSIs or primer mismatches [5,26,53]. Detection of CBSIs in asymptomatic plants is a common feature of the CBSD disease phenomenon, as symptoms are often cryptic and sensitive to seasonal variation [74]. The detection rate for CBSIs in asymptomatic plants of 16% demonstrates that the true levels of infection are significantly greater than those measured by using visual assessment, although this would not represent a large increase in the incidence level determined from symptoms as it would still result in an overall true incidence of less than 20%. These results suggest that although CBSD continues to be an important cassava production constraint in western Kenya, its status is moderate and relatively stable.

The number of adult whiteflies in 2023 was nine times higher compared to those in 2022, when four outlier fields with variety MM96/2480 were excluded from the averages, while nymphs were three times higher, suggesting that conditions prevailing during the 2023 season were favourable for whitefly population build-up. This was anticipated, as weather conditions are known to be more favourable for *B. tabaci* whiteflies on cassava during the hottest time of the year in February and March. Bungoma, which had the fewest adult whiteflies in 2022, had the highest number in 2023, suggesting the high variability of whitefly numbers depending on the prevailing season, which in turn could influence CMD and CBSD epidemics. In this study, the high number of whiteflies in 2023 coincided with a higher proportion of CMD infection attributed to whitefly transmission (50.6%) compared to 2022, which had 18%. The association between whitefly abundance and cassava virus spread is well documented [75,76], and the importance of seasonal effects on whitefly abundance and CBSD spread has been clearly documented for coastal Tanzania, where high whitefly abundance and rapid CBSD spread were associated with planting in the short rainy season, in contrast to much lower whitefly abundances and less CBSD spread for plantings during the long rainy season [73]. The average number of adult whiteflies (47) and nymphs (62) reported in this study, excluding the four outlier fields in Homa Bay, is very high compared to reports from other recent regional surveys; for example, 0.9 adults and 5.2 nymphs in Rwanda [65], 4.7 (2018) and 1.8 (2016) adult whiteflies in eastern DRC [25], 0.1 to 15.9 adults in Benin [70], 1.8 adults in Comoros Islands [26], and 0.6 adults in Zambia [69]. The abundances of whitefly adults on improved varieties were approximately double those on local varieties. The abundances of *B. tabaci* on MM96/4271 were typical of this pattern. In addition, however, two fields in Homa Bay with the improved variety MM96/2480 had an unusually high number of whiteflies. The whitefly abundances were some of the highest recorded on cassava for two of the four fields where the variety was recorded (3280 and 1961), with a highest single plant count of 7000, although the two other fields had much lower abundances of 89 and 10. Further research will be required to determine whether these extreme abundances are the result of the ultra-suitability of the variety for cassava *B. tabaci* or the consequence of specific and unusual micro-environmental conditions.

Several other studies have reported a higher number of whiteflies on improved varieties compared to local landraces; for example, three times higher for both whiteflies and nymphs in Rwanda [65], two times higher nymph means in Uganda [66], and high numbers on improved varieties compared to local landraces [77]. MM96/4271 was among the varieties hosting higher mean numbers of whiteflies in a study that evaluated resistance among 23 varieties selected from East and Southern Africa [78]. It was notable, however, that these differences in abundance did not translate into differences in the incidence of virus disease, suggesting that the improved varieties must have generally higher levels of virus resistance than their local equivalents.

The cassava *B. tabaci* mitotypes that were detected included SSA1-SG1, SSA1-SG2, SSA1-SG1/SG2 and SSA2. These findings are consistent with previous studies for samples from western Kenya [29,38,39]. The predominant mitotype was SSA1-SG1, with 64.4% in 2017 and 70% in 2022/2023, which is consistent with most previous studies in Eastern Africa [29,38,39,79,80] except for South Sudan, where SSA2 was the most frequently encountered cassava *B. tabaci* mitotype [81]. The occurrence of SSA2 and SSA1-SG2 appeared to reciprocally reduce or increase depending on the time of sampling; in 2017, SSA2 accounted for 28.8% and SSA1-SG2 for 5.5%, while in 2022/2023, their proportions reversed to 3% and 26.5%, respectively. The fluctuating occurrence and even absence of SSA2 in samples collected from regions in Uganda and Kenya has been reported in previous studies [29,39,55,82,83]. A fourteen-year trend of SSA2 (1997–2010) reported high frequency during the period of 1997–1999, moderate occurrence in 2000–2001 and very low frequency in 2004–2010 [29]. So far, no explanation has been suggested for this SSA2 trend. KASP SNP genotyping revealed SSA-ECA to be the most frequently occurring haplogroup, accounting for 72% in 2017 and 93.6% in 2022/2023. This is reported to be the most widespread haplogroup across large parts of East and Central Africa [38,39,40,84]. SSA-ECA is dominant in areas severely affected by CMD and CBSD epidemics, and its persistent presence in high numbers in western Kenya is an indication that the region remains under continual threat of virus epidemics. KASP failed to clearly designate 6% of the samples in any of the known six haplogroups. This diagnostic tool was developed on a limited number of samples and this failure could be attributed to primer mismatches and provides an indication that there is a need for continuous optimisation using diverse samples.

No obvious relationship was apparent between the whitefly numbers and the incidence of CMD and CBSD. In 2023, the whitefly numbers were nine times higher yet the incidence of CMD was 10.1% and CBSD 4.1% compared to 26.4% and 6.4% in 2022. Furthermore, the proportion of plants that were scored as whitefly-infected was unchanged between the years (4.4%). The lack of relationship between high whitefly numbers and virus incidence could be attributed to several factors. Firstly, there is a lag between adult whitefly population abundances and the expression of symptoms resulting from the virus transmission that they cause, since there is a latent period for symptom expression of approximately one month for both CMD and CBSD [85,86], and secondly, the improved varieties where whiteflies were particularly abundant are also resistant or tolerant to CMD and CBSD [10]. The high incidence of CMD in 2022 compared to 2023 is attributed to a higher incidence of infected cuttings in 2022, which is an indication of a lack of clean planting material. In a survey carried out in coastal Kenya, it was found that 82.5% of the farmers recycled planting material from the previous crop, 67.5% obtained material from neighbours or sourced it from other regions, 11% obtained planting material from research organisations, 5.3% bought from a market and only 2.5% sourced clean material every season [5]. Even though the virus incidence levels were not very high, the large number of fields with super-abundant whitefly populations (> 100 adults/five top leaves) in 2023 (28%) compared to 2022 (5%) indicates that whiteflies pose a threat as a physical pest in seasons in which they occur in large numbers. Whitefly damage alone can cause up to 50% yield loss under severe infestation [33]. In a study evaluating the efficacy of cutting dipping in insecticides against whiteflies in cassava under high whitefly population and virus inoculum pressure, Flupyradifurone (Sivanto SL 200) reduced the whitefly numbers by 41% for adults and 65% of nymphs, and the CMD incidence was 34% lower than in the untreated control [87]. Control of whiteflies using cutting dipping in insecticides contributed to a 49% root yield increase, which clearly demonstrated the potential benefit of whitefly control [87].

Cassava green mite was widespread across all the counties, with a prevalence of 95% and incidence in the range of 49.6 to 91.3%. These findings indicate that CGM, which was previously under control, probably due to the combined action of predatory mites and rainfall, could be re-emerging as a serious pest due to erratic rainfall patterns that have led to prolonged drought conditions in many cassava growing regions [47]. Drought favours rapid establishment of CGM and could also reduce the efficiency of predatory mites in managing this pest [42,47]. Farmers in Rwanda ranked poor-quality planting material and unpredictable rains/drought as the major challenges affecting cassava production [88]. The response of varieties in the current study shows that all the varieties are prone to CGM infestation, as some that were found to have low incidence and severity in 2022 were found to have high incidence and severity in 2023, and vice versa. However, the most predominant variety, MM96/4271, had a lower severity of 2.4 compared to the overall average of 2.7. Considering that the fields sampled in this study were 3 to 6 months old, the severity of the CGM damage is likely to have increased as the plants matured further. An increased frequency of unpredictable weather conditions is expected to be a consequence of anthropogenic climate change. Although cassava has been shown to be the most adaptable of the major staple crops to the anticipated effects of climate change [6], there will be changes in interactions with the major pests and diseases, and research will be required to determine the most appropriate and effective ways in which to respond to these changes. This will be particularly important for CGM, where control has depended on a delicate tri-trophic balance between the pest, exotic and indigenous natural enemies, as well as the cassava host plant.

## 5. Conclusions

CMD, CBSD, *Bemisia* whiteflies, and cassava green mite continue to pose a significant threat to cassava production in western Kenya. Surveys conducted in the two main cassava growing seasons of western Kenya revealed some of the highest cassava whitefly abundances ever reported, with average counts of > 3000 adult whiteflies per plant recorded from one location. In spite of these extraordinarily high vector populations, the incidences of CMD and CBSD were moderate to low in both seasons. This seems to be in large part due to the high level of adoption of improved virus-resistant varieties (60%). Although this represents an important achievement, which is likely delivering significant benefits to the region’s farmers through increased yields, the sustained abundance of whitefly vectors does represent an on-going threat, both since new virus strains may emerge to which the current varieties lack resistance and also since super-abundant whitefly populations can themselves cause physical damage to cassava crops. Furthermore, CGM damage is widespread, moderate to severe, and may be exacerbated by the effects of climate change. These points highlight the need for on-going efforts to enhance cassava pest and disease control, which should include the sustained management of cassava viruses through breeding for resistance and clean seed delivery, the deployment of whitefly control strategies, and a re-assessment of biological control tactics for CGM control with a view to assuring their robustness to the effects of climate change.

## Figures and Tables

**Figure 1 viruses-16-01469-f001:**
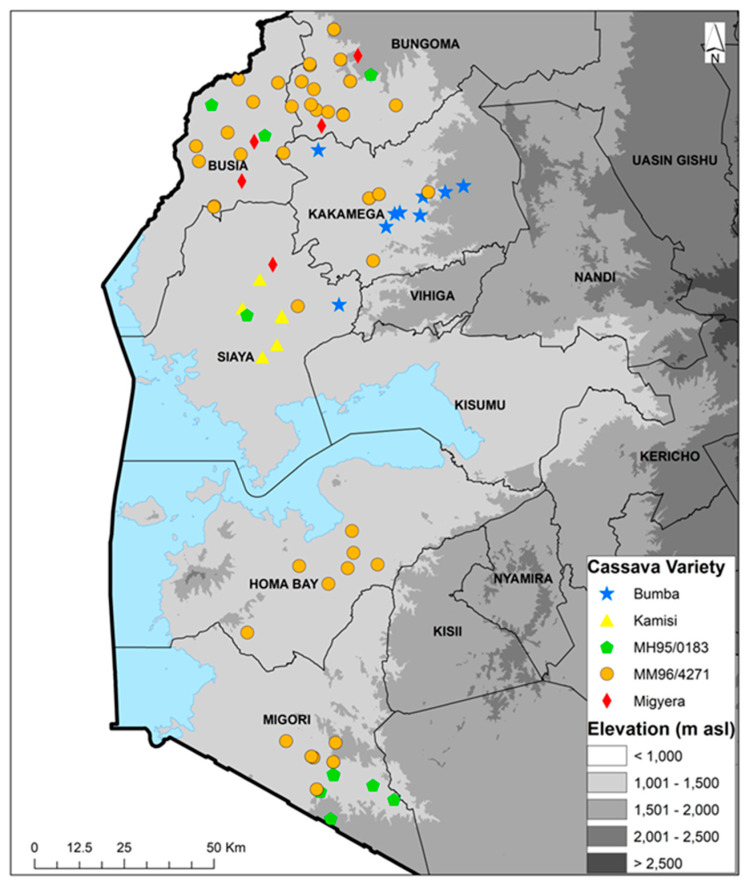
Geographic distribution of the top five cassava varieties cultivated in 2022 and 2023 in six counties of western Kenya.

**Figure 2 viruses-16-01469-f002:**
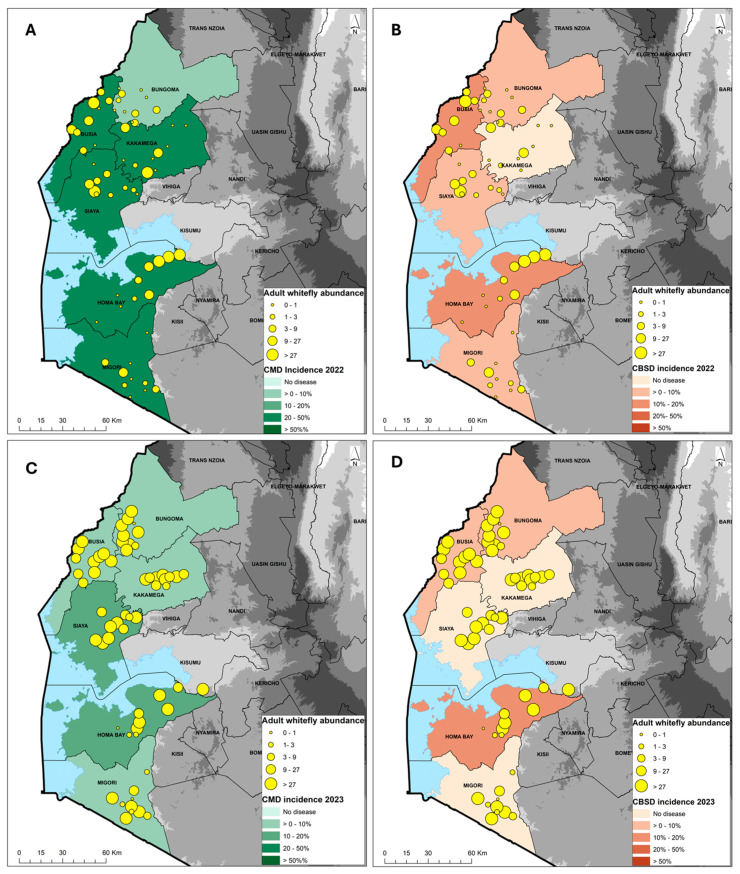
Geographic distribution of cassava mosaic disease (CMD) and cassava brown streak disease (CBSD) in 2022 (**A**,**B**) and 2023 (**C**,**D**) in relation to the *Bemisia tabaci* whitefly abundance across six counties in western Kenya.

**Figure 3 viruses-16-01469-f003:**
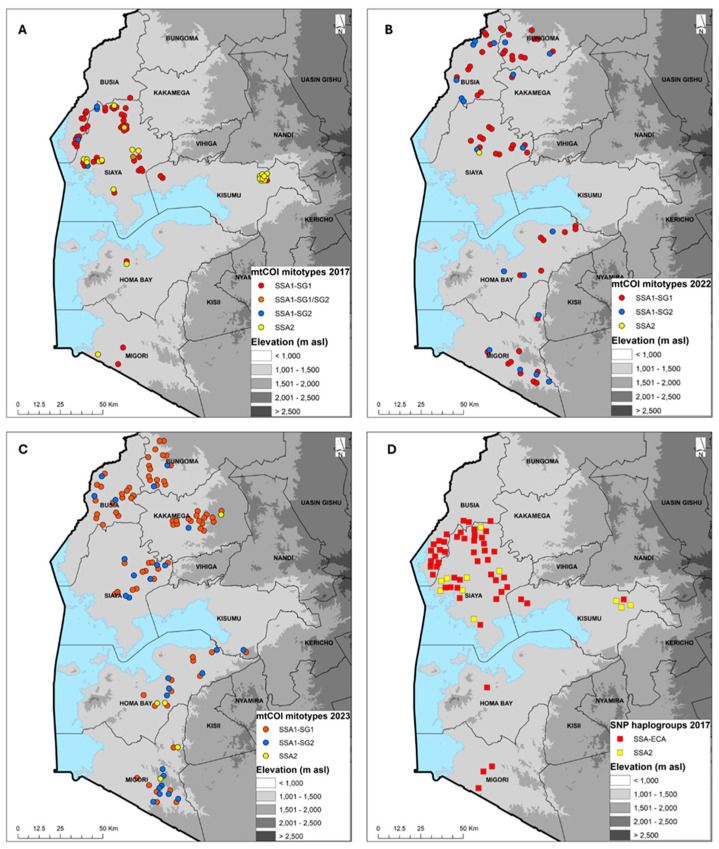
Geographic distribution of cassava-colonizing *Bemisia tabaci* whiteflies in counties surveyed in western Kenya based on mtCOI sequencing (2017 (**A**), 2022 (**B**) and 2023 (**C**) and KASP SNP genotyping 2017 (**D**)).

**Figure 4 viruses-16-01469-f004:**
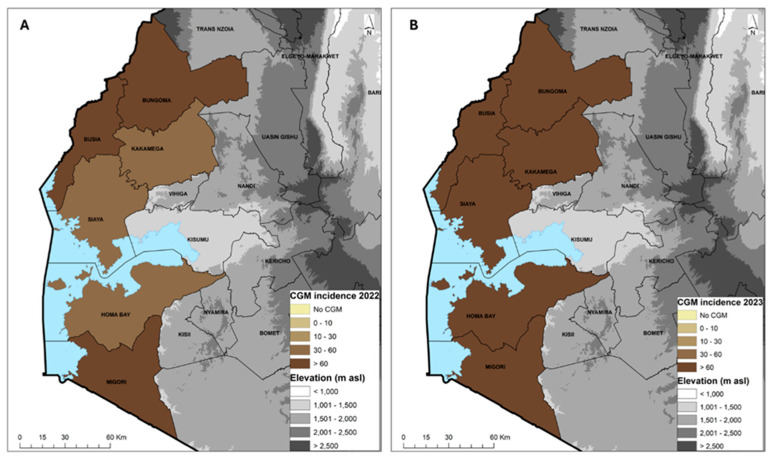
Geographic distribution of the cassava green mite (CGM) incidence in 2022 (**A**) and 2023 (**B**) across six counties in western Kenya.

**Table 1 viruses-16-01469-t001:** Incidence, prevalence, and severity of cassava virus diseases and abundance of the whitefly *Bemisia tabaci* for varieties surveyed in Western Kenya, October 2022.

County	Variety	Sites	*B. tabaci* Adults	*B. tabaci* Nymphs	Leaf CBSD Sev. Score	Leaf CBSD Inc. (%)	CMD Sev. Score	CMD Inc. (%)	CBSD Prev. (%)	CMD Prev. (%)
Bungoma	MH95/0183	1	0.4	2.8	-	0.0	2.8	13.3		
	Migyera (TMS 30572)	1	0	3.4	-	0.0	3.5	6.7		
	MM96/4271 (NASE 14)	7	3.8	11.8	2.0	0.5	2.5	3.8	10.0	70.0
	MM98/5280	1	0	6.2	-	0.0	3.3	30.0		
Busia	Fumba Chai	1	0.5	2.2	-	0.0	3.4	70.0		
	Magana	1	0.1	0	2.1	30.0	3.5	56.7		
	MM96/4271	6	26.9	35.5	2.8	4.4	3.3	5.0	40.0	60.0
	Red	1	18.7	48.2	2.7	96.7	-	0.0		
	Yellow	1	6.9	7.8	-	0.0	3.7	80.0		
Homa Bay	Adhiambo Lera	1	5.0	90.2	2.0	3.3	3.2	23.3		
	MM96/2480	4	1334.9	727.6	3.0	0.8	2.9	25.0	40.0	70.0
	MM96/4271	3	66.8	8.4	3.1	32.2	-	0.0		
	Serere	2	1.2	1.8	-	0.0	3.9	61.7		
Kakamega	Adhiambo Lera	1	0	4.8	-	0.0	2.7	93.3		
	Bumba	4	10.2	168.8	-	0.0	3.1	69.2		
	MM96/4271	1	72.2	248.8	-	0.0	2.8	30.0	0.0	70.0
	MM96/5280	1	0.1	1.2	-	0.0	-	0.0		
	MM96/7151	2	1.5	26.4	-	0.0	-	0.0		
	Serere	1	0	56.8	-	0.0	3.1	26.7		
Migori	Merry-go-round	3	0.4	1.4	-	0.0	3.9	36.7		
	MM95/0183	3	1.6	4.5	3.0	1.1	4.0	1.1		
	MM96/0686	1	6.3	0	-	0.0	3.8	26.7	10.0	60.0
	MM96/4271	2	9.9	5.1	-	0.0	-	0.0		
	Sudhe	1	0	0.6	-	0.0	3.8	100		
Siaya	Bumba	1	2.1	18	-	0.0	3.1	26.7		
	Kamisi	2	34.6	51.1	-	0.0	3.5	46.7		
	MH95/0183	1	1.1	0.2	-	0.0	4.0	3.3	20.0	80.0
	Migyera	1	3.4	13	-	0.0	-	0.0		
	MM96/4271	1	0.7	4.2	-	0.0	-	0.0		
	Nyakatanegi	3	6.3	12.5	2.9	40.0	3.8	57.7		
	Nyanjaga	1	1.4	8.4	-	0.0	3.1	60.0		

CBSD: cassava brown streak disease; CMD: cassava mosaic disease; Sev.: severity; Inc.: incidence; Prev.: prevalence; Av.: average; -: no foliar symptoms observed.

**Table 2 viruses-16-01469-t002:** Incidence, prevalence, and severity of cassava virus diseases and abundance of the whitefly *Bemisia tabaci* for varieties surveyed in Western Kenya, March 2023.

County	Variety	Sites	*B. tabaci* Adults av.	*B. tabaci* Nymphs av.	Leaf CBSD Sev. Score	Leaf CBSD Inc. (%)	CMD Sev. Score	CMD Inc. (%)	CBSD Prev. (%)	CMD Prev. (%)
Bungoma	Migyera	1	93.1	60.0	-	0.0	-	0.0		
	MM96/4271	8	176.5	92.3	2.1	6.3	2.7	3.0		
	Nylon	1	202.0	12.4	-	0.0	2.0	6.7	30.0	50.0
Busia	Matuja	1	14.6	20.2	-	0.0	2.0	13.3		
	MH95/0183	2	111.2	19.4	-	0.0	3.1	23.3		
	Migyera	2	110.0	44.0	-	0.0	2.5	8.3		
	TMS I92/0067	1	123.3	99.2	-	0.0	-	0.0		
	MM96/4271	4	181.1	122.1	2.5	20.8	3.0	0.8	20.0	70.0
Homa Bay	MM96/4271	4	41.4	74.4	2.1	25.0	2.5	1.6		
	Nyakatanegi	1	28.5	224.0	3.5	83.3	3.6	33.3		
	Nyakichagi	4	40.4	29.4	-	0.0	3.3	16.7		
	Unknown	1	69.4	324.0	3.0	13.3	3.6	90.0	40.0	80.0
Kakamega	Bumba	4	20.6	62.3	-	0.0	2.6	5.8		
	Fumba Chai	1	56.1	234.0	-	0.0	3.2	20.0		
	MM96/4271	3	95.5	294.0	-	0.0	-	0.0		
	Unknown	2	47.7	192.2	-	0.0	3.4	15.0	0.0	50.0
Migori	MH95/0183	3	20.7	30.7	-	0.0	3.5	8.8		
	MH96/0686	1	1.1	0.0	-	0.0	-	0.0		
	MM96/4271	4	53.1	51.3	-	0.0	2.9	8.3		
	MM98/3567	1	282.1	183.0	-	0.0	-	0.0		
	Nyanchagi	1	9.0	10.8	-	0.0	-	0.0	0.0	40.0
Siaya	Adhiambo Lera	1	65.3	98.2	-	0.0	3.5	86.7		
	Kamisi	4	72.9	64.6	-	0.0	3.2	5.8		
	MM96/4271	1	89.1	114.8	-	0.0	-	0.0		
	MM97/2480	1	67.9	298.0	-	0.0	3.0	13.3		
	Nylon	2	20.9	39.7	-	0.0	2.5	23.3		
	Sudhe	1	3.7	4	-	0.0	3.0	3.3	0.0	70.0

CBSD: cassava brown streak disease; CMD: cassava mosaic disease; Sev.: severity; Inc.: incidence; Prev.: prevalence; Av.: average; -: no foliar symptoms observed.

**Table 3 viruses-16-01469-t003:** Real-time RT-qPCR testing results for cassava brown streak ipomoviruses in cassava leaves collected from field plants in Western Kenya, October 2022.

Real-Time RT-qPCR Testing for CBSIs in Symptomatic Samples					
County	Number of samples	CBSV positive	UCBSV positive	CBSIs negative	% symptomatic positive
Bungoma	10	7	0	3	70.0
Busia	12	11	0	1	91.6
Homa Bay	12	7	6	2	83.3
Kakamega	0	0	0	0	-
Migori	1	1	0	0	100
Siaya	8	3	6	1	87.5
Real-time RT-qPCR testing for CBSIs in asymptomatic samples					
County	Number of samples	CBSV positive	UCBSV positive	CBSIs negative	% asymptomatic positive
Bungoma	90	10	0	80	11.0
Busia	87	27	3	57	34.5
Homa Bay	88	17	3	68	22.7
Kakamega	100	1	0	99	1.0
Migori	99	8	1	91	8.1
Siaya	92	13	5	75	18.5

**Table 4 viruses-16-01469-t004:** Whitefly *Bemisia tabaci* and *Bemisia afer* distribution in Western Kenya in 2017, 2022 and 2023 (mtCOI sequencing).

	SSA1-SG1	SSA1-SG2	SSA1-SG1/SG2	SSA2	*B. afer*
2017	47/73 (64.4%)	4/73 (5.5%)	1/73(1.4%)	21/73(28.8%)	0/73 (0%)
2022	52/73 (71.2%)	20/73 (27.4%)	0/73 (0%)	1/73 (1.4%)	0/73 (0%)
2023	78/112 (69.6%)	27/112 (24.1%)	0/112(0%)	5/112 (4.5%)	2/112 (1.8%)

**Table 5 viruses-16-01469-t005:** Whitefly cassava *Bemisia tabaci* distribution in Western Kenya in 2017, 2022 and 2023 (KASP SNP genotyping).

Year	SSA-ECA	SSA2	Others
2017	59/82 (72.0%)	13/82 (15.8%)	10/82 (12.2%)
2022	103/111 (92.8%)	2/111 (1.8%)	6/111 (5.4%)
2023	219/232 (94.4%)	11/232 (4.7%)	2/232 (0.9%)

**Table 6 viruses-16-01469-t006:** Comparison of cassava green mite (CGM) severity and incidence across six counties in western Kenya in October 2022 and March 2023.

County	October 2022			March 2023		
	CGM Score	CGM Incidence (%)	CGM Prevalence (%)	CGM Score	CGM Incidence (%)	CGM Prevalence (%)
Bungoma	2.4	68.7	100	2.6	79.7	100
Busia	2.5	68.3	100	2.8	91.3	100
Homa Bay	2.9	55.3	100	2.4	76.7	80
Kakamega	2.2	64.8	90	2.8	91.3	100
Migori	3.0	82.7	100	2.6	64.0	100
Siaya	2.9	49.6	80	2.8	69.3	90
Average	2.7	65.4	95	2.7	78.9	95

**Table 7 viruses-16-01469-t007:** Comparison of CGM severity and incidence in the four most commonly occurring varieties in Western Kenya in October 2022 and March 2023.

Variety	October 2022		March 2023	
	CGM Score	CGM Incidence (%)	CGM Score	CGM Incidence (%)
Bumba	2.3	53.3	3.2	98.3
MH95/0183	3.3	83.3	2.7	49.3
MM96/4271	2.4	83.3	2.4	77.9
Migyera	2.8	53.3	3.2	88.8

## Data Availability

Data are available upon request.

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
