# Peer review of "Occurrence and Distribution of Major Cassava Pests and Diseases in Cultivated Cassava Varieties in Western Kenya"

_viruses, 2024, doi:10.3390/v16091469_

Round 1

Reviewer 1 Report

Comments and Suggestions for Authors

The paper by Wosula et al. provides a robust and detailed examination of the occurrence and distribution of cassava pests and diseases in Western Kenya.

The research is commendable for its wide geographical coverage 6 counties including 120 fields, which enhances the reliability of the findings. The incorporation of advanced molecular techniques, such as KASP SNP genotyping, significantly strengthens the scientific rigor, particularly in the accurate identification of whitefly populations. Additionally, the study offers valuable insights into the prevalence of improved cassava varieties, highlighting their resistance to Cassava Mosaic Disease (CMD), relative tolerance to Cassava Brown Streak Disease (CBSD), and the coverage of Cassava Green Mite (CGM) and whitefly types.

The materials and methods section is well-structured and detailed, providing a solid foundation for the results. The results are clearly presented, with strong statistical data that effectively highlight pest and disease prevalence across different cassava varieties and regions. The discussion appropriately addresses the broader implications of the findings, including the importance of adopting improved cassava varieties and the need for integrated pest management strategies. Considering the breeding for resistance, distributing clean seed, implementing whitefly control strategies, and reassessing CGM biological control measures to ensure their effectiveness taking into the account climate change

The writing is generally clear, with appropriate use of scientific language.

Overall, the paper makes valuable contributions to the field of cassava pest and disease management and is highly recommended for publication.

Reviewer 2 Report

Comments and Suggestions for Authors

the article is very well writen and as a good structure. Only one remark that will improve even more the article in my opinion is to put in the methods part how you calculate the incidence and prevalence  that is show in the results (line 272) this because you state the scales used but not how you reach these two important parameters.

Author Response

Reviewer 1

The article is very well written and as a good structure. Only one remark that will improve even more the article in my opinion is to put in the methods part how you calculate the incidence and prevalence that is show in the results (line 272) this because you state the scales used but not how you reach these two important parameters.

Response: Details on calculation of incidence and prevalence have been included in the manuscript.

Reviewer 3 Report

Comments and Suggestions for Authors

Everlyne et al. surveyed 120 fields to determine the status of major cassava pests and diseases in six counties in western Kenya, a leading region in cassava production. This study is of great interest in proposing management recommendations to farmers. However, there are some minor issues to address. The authors could group the samples collected in Groups A and B instead of using 2022 and 2023 or October and March. Providing more information about the improved and local varieties will be better.

Title: It would be great to mention that cassava varieties were also surveyed. They also reported that most of the varieties recorded are improved.

The abstract is not well organized. Please refer to the instructions for the journal's authors. Especially missing a background and conclusions of the main findings. For instance, from this study, we can see that the improved varieties could reduce the incidence of CMD, but CBSD remains a severe threat that might lack control strategies. Variety MM96/4271 is susceptible to CBSD that the wild types... Furthermore, it would be of great interest if the authors could point out recommendations for managing these diseases from this study.

Lines 13-16: The authors mentioned they had surveyed to determine the prevalence, incidence, and severity of cassava mosaic disease (CMD) and cassava brown streak disease (CBSD), whitefly numbers, and incidence of cassava green mite (CGM) in six counties of western Kenya. Then, they mentioned that they have recorded 29 varieties. They also surveyed the cassava varieties in their study.

Lines 18-20, 25-27, and 29-31: The difference between the two batches of collection is mainly the climate. One was during a short rainy season, and the other was during a long rainy period. These data imply that short rainy seasons increase the severity of CMD, CBSD, and whitefly; however, relatively reduced the prevalence and incidence of CMD and whitefly but constant in CBSD. What could explain this? Besides, have the authors recorded the main temperatures between these two groups of samples?

Line 31, Full name of mtCOI

Line 33: full name of CGM

Line 149-150: Does it mean one predominant variety was assessed?

Author Response

Reviewer 2

Everlyne et al. surveyed 120 fields to determine the status of major cassava pests and diseases in six counties in western Kenya, a leading region in cassava production. This study is of great interest in proposing management recommendations to farmers. However, there are some minor issues to address. The authors could group the samples collected in Groups A and B instead of using 2022 and 2023 or October and March. Providing more information about the improved and local varieties will be better.

Response: The year and month of sampling are used to guide on time of the year surveys were carried out as season impacts whitefly numbers and virus disease epidemics in cassava due to influence of rain and temperature. We therefore prefer to have month and year.

Title: It would be great to mention that cassava varieties were also surveyed. They also reported that most of the varieties recorded are improved.

Response: Varieties added in the title

The abstract is not well organized. Please refer to the instructions for the journal's authors. Especially missing a background and conclusions of the main findings. For instance, from this study, we can see that the improved varieties could reduce the incidence of CMD, but CBSD remains a severe threat that might lack control strategies. Variety MM96/4271 is susceptible to CBSD that the wild types... Furthermore, it would be of great interest if the authors could point out recommendations for managing these diseases from this study.

Response: The abstract has been written to include background and conclusion

Lines 13-16: The authors mentioned they had surveyed to determine the prevalence, incidence, and severity of cassava mosaic disease (CMD) and cassava brown streak disease (CBSD), whitefly numbers, and incidence of cassava green mite (CGM) in six counties of western Kenya. Then, they mentioned that they have recorded 29 varieties. They also surveyed the cassava varieties in their study.

Response: included details on survey of cultivated cassava varieties

Lines 18-20, 25-27, and 29-31: The difference between the two batches of collection is mainly the climate. One was during a short rainy season, and the other was during a long rainy period. These data imply that short rainy seasons increase the severity of CMD, CBSD, and whitefly; however, relatively reduced the prevalence and incidence of CMD and whitefly but constant in CBSD. What could explain this? Besides, have the authors recorded the main temperatures between these two groups of samples?

Response: There is a trend in CBSD, CMD and whitely abundance fluctuating depending on the season. Rainfall and low temperatures reduce whitefly activity which contributes to reduced virus epidemics though this is also dependent on quality of cassava cuttings planted. For example, in this study the incidence of CMD is high and whitefly numbers are low, while in 2023 it is the reverse. Whitefly borne CMD incidence for both years is 4% meaning the higher incidence in 2022 is attributed to planting of infected material. This explanation is in the discussion section              line 543-556

We have also highlighted the point that whitefly CMD infection in 2023 was greater as a proportion of total CMD infection in 2023 than it was in 2022. Lines 275-278; 299-302

This study did not record temperature.

Line 31, Full name of mtCOI

Response: This line has been deleted to comply with the word limit after adding the background and conclusion in the abstract.

Line 33: full name of CGM

Response: Thanks. This was abbreviated in line 15: now reads as line 17:

Line 149-150: Does it mean one predominant variety was assessed?

Response: To avoid confusion, the phrase “For each selected field…” has been added in line 150:
